# Transforming Growth Factor Beta and Alveolar Rhabdomyosarcoma: A Challenge of Tumor Differentiation and Chemotherapy Response

**DOI:** 10.3390/ijms25052791

**Published:** 2024-02-28

**Authors:** Bhavya Bhushan, Rosa Iranpour, Amirmohammad Eshtiaghi, Simone C. da Silva Rosa, Benjamin W. Lindsey, Joseph W. Gordon, Saeid Ghavami

**Affiliations:** 1Department of Human Anatomy and Cell Science, Max Rady Faculty of Health Sciences, University of Manitoba, Winnipeg, MB R3E 0W2, Canada; bhavya.bhushan@mail.mcgill.ca (B.B.); iranpou1@myumanitoba.ca (R.I.); eshtiaga@myumanitoba.ca (A.E.); simone.dasilvarosa@umanitoba.ca (S.C.d.S.R.); benjamin.lindsey@umanitoba.ca (B.W.L.); joseph.gordon@umanitoba.ca (J.W.G.); 2Department of Anatomy and Cell Biology, School of Biomedical Sciences, Faculty of Science, McGill University, Montreal, QC H3A 0C7, Canada; 3Department of Biomedical Engineering, University of Manitoba, Winnipeg, MB R3T 2N2, Canada; 4Research Institute of Oncology and Hematology, Cancer Care Manitoba, University of Manitoba, Winnipeg, MB R3E 0V9, Canada

**Keywords:** rhabdomyosarcoma, combination therapy, all-trans retinoic acid, TGF-beta1

## Abstract

Alveolar rhabdomyosarcoma (ARMS), an invasive subtype of rhabdomyosarcoma (RMS), is associated with chromosomal translocation events resulting in one of two oncogenic fusion genes, *PAX3-FOXO1* or *PAX7-FOXO1*. ARMS patients exhibit an overexpression of the pleiotropic cytokine transforming growth factor beta (TGF-β). This overexpression of TGF-β1 causes an increased expression of a downstream transcription factor called SNAIL, which promotes epithelial to mesenchymal transition (EMT). Overexpression of TGF-β also inhibits myogenic differentiation, making ARMS patients highly resistant to chemotherapy. In this review, we first describe different types of RMS and then focus on ARMS and the impact of TGF-β in this tumor type. We next highlight current chemotherapy strategies, including a combination of the FDA-approved drugs vincristine, actinomycin D, and cyclophosphamide (VAC); cabozantinib; bortezomib; vinorelbine; AZD 1775; and cisplatin. Lastly, we discuss chemotherapy agents that target the differentiation of tumor cells in ARMS, which include all-trans retinoic acid (ATRA) and 5-Azacytidine. Improving our understanding of the role of signaling pathways, such as TGF-β1, in the development of ARMS tumor cells differentiation will help inform more tailored drug administration in the future.

## 1. Rhabdomyosarcoma Overview

Rhabdomyosarcoma (RMS) is a highly recurrent pediatric soft-tissue sarcoma accounting for approximately 50% of childhood cancer cases, with a higher incidence in males than females, with a male/female ratio of 1.4:1 [1,2,3]. RMS is considered as a high-grade, malignant neoplasm wherein the cells resemble myoblasts, express myogenic markers, and are suspected to be derived from muscle progenitor cells [3,4,5]. The expression of myogenic factors such as MyoD strongly supports the hypothesis that RMS originates from a mesenchymal cell lineage that gives rise to myoblasts [1,4,5]. In addition to occurring within the skeletal muscle of the extremities, due to its totipotent origin, RMS can also occur in regions such as the head, neck, genitourinary organs, abdomen, and bile ducts [2,6,7]. According to the World Health Organization (WHO), there are four classifications of RMS, namely alveolar, embryonal, spindle cell/sclerosing, and pleomorphic RMS [7] (Figure 1). However, histologically alveolar (~60–70% of the cases) and embryonal (~20–30% of the cases) RMS are considered to be the two major subtypes [7,8]. Due to its high tendency to metastasize, alveolar rhabdomyosarcoma (ARMS) is considered an invasive subtype largely found at the extremities of adolescents [9]. Its name is derived from its histological features, as it exhibits a fibrovascular septum of connective tissue with neoplastic cells attached, vaguely reminiscent of the alveolar spaces in the lungs [2,10]. The cells themselves can be characterized as uniformly polygonal with an oval/round nucleus that is hyperchromatic [2,11].

## 2. Tumor Growth and Differentiation in ARMS

### 2.1. PAX3/7 and FOXO1

Epithelial to mesenchymal (EMT) is a multi-step process. During EMT, epithelial cells are reprogrammed into a mesenchymal phenotype, decreasing cell-to-cell adhesion, cytoskeleton remodeling, basement membrane invasion, and acquisition of motility, and causing a lack of cell polarity (Sannino et al., 2017). This results in normally cohesive cells shifting from their rigid epithelial organization and metastasizing to other loci, contributing to cancer progression and drug resistance [12]. The commitment of cells to the EMT program is orchestrated by specific transcription factors, therefore making transcription factors a major focus of cancer research.

Significantly, ~80% of the ARMS cases manifest due to a chromosomal translocation between the paired box protein *PAX3* gene found on chromosome 2 or the *PAX7* gene found on chromosome 1, and the forkhead box protein *FOXO1* gene on chromosome 13. This results in fusion genes (*PAX 3/7-FOXO1*) that are considered major drivers of oncogenic activity [7,13] (Figure 2). The N-terminal DNA binding domain of *PAX3* or *PAX7* is fused to the C-terminal transactivation domain of *FOXO1* [5,14].

Transcription factors, known as myogenic factors, not only regulate the physiological process of muscle formation (i.e., myogenesis), but also play an important role in pathologic myogenic differentiation. Accordingly, aberrations within these pathways can lead to the development of RMS [15,16]. Previous studies suggest that the development of RMS correlates with a differentiation defect in either stem cells or early progenitor cells such as mesenchymal stem cells [15,17]. ARMS not only has the presence of the PAX3/PAX7 and FOXO1 fusion genes, but also displays increased levels of MET, a receptor tyrosine kinase (RTK) known to be linked with the metastatic potential of RMS cells [15,18]. Furthermore, recent studies have shown that aggressive ARMS tumors exhibit high expressions of SNAIL, which coincidently has a positive correlation to PAX3/7-FOXO1 (15, 19–21). SNAIL has thus become a prominent research focus as it plays a central role in EMT [15,19].

SNAIL is a zinc finger transcription factor that usually acts as a gene repressor, making it a crucial regulator of ARMS growth [20,21]. It is known to be a master regulator of growth and metastasis as it involves the induction of EMT by binding to E-box sequences in the promoter region on genes, affecting the EZRIN cytoskeleton protein and AKT serine/threonine kinase levels, respectively [16,20,21]. In ARMS, the PAX3/7-FOXO1 fusion gene not only alters the myogenic program and maintains the proliferative state while blocking terminal differentiation, but it also results in an overexpression of TGF-β [22,23]. This overexpression of TGF-β results in an upregulation of its two major downstream pathways, further allowing SNAIL to promote EMT (Figure 3). Additionally, SNAIL also regulates the microRNA transcriptome either directly or indirectly by binding their promoter or regulatory regions in RMS cells [16,19,21]. To this end, gene ontology analysis has revealed that the SNAIL–miRNA axis does in fact play a role in differentiation, the reorganization of the actin skeleton, and migration [16,19,20,21].

### 2.2. Transforming Growth Factor-β

In addition to the PAX 3/7-FOXO1 fusion gene, ARMS cells also exhibit an increased expression of TGF-β family proteins, in particular TGF-β1 [24,25]. TGF-β1, a member of the transforming growth factor (TGF) superfamily, is a pleiotropic cytokine that is secreted by fibroblasts and epithelial cells [26,27,28]. It is encoded by 33 genes in mammals with the capability to function as both a homo- and heterodimer [27,29]. In addition to the function of TGF-β in cellular development, different members of the TGF-β superfamily are intensely studied for their widespread role in various diseases, including cancer [29,30]. TGF-β is known to strongly inhibit the proliferation of many cell types, including endothelial, epithelial, and immune cells, while playing a prominent role in controlling the differentiation of cell lineages during development [29,31,32]. Additionally, TGF-β family proteins are involved in other cellular functions such as promoting/protecting against cell death, cell motility, and invasion [29,31,33].

As shown in Figure 4, the TGF-β receptor complex transduces signals to regulate context-dependent transcription through two major downstream signaling pathways: (1) SMAD pathways (canonical) and (2) non-SMAD pathways (non-canonical) (32). The canonical TGF-β pathway transfers signals via SMAD2/3, and once phosphorylated by TβR I, it regulates gene transcription by translocating into the nucleus (32). On the contrary, the non-canonical pathway refers to the activation of downstream cascades by TGF-β via phosphorylation, acetylation, protein–protein interactions, ubiquitination, and sumoylation, as well as involving the MAPK/ERK kinases [32].

TGF-β plays a dual role in cancer [23] (Figure 5). Under normal expression, TGF-β has a tumor suppressor effect and promotes differentiation. Conversely, when overexpressed, as in the case of ARMS, TGF-β is known to decrease differentiation and inhibit myogenic differentiation [23]. The inhibition of myogenic differentiation is considered a negative effect, as undifferentiated cells are more resistant to chemotherapy. This supports the notion that ARMS is not only an invasive subtype but also highly resistant to therapy [23]. Overexpression also results in an over-activation of the downstream cascades, resulting in the induction of EMT and cancer progression.

Thus, using chemotherapy agents that both increase differentiation and kill ARMS tumor cells could be advantageous for better treatment of this malignant tumor.

## 3. ARMS Chemotherapy Drugs and Their Impact on Tumor Cell Differentiation

Currently, multiple approaches consisting of a combination of surgery, chemotherapy, and/or radiotherapy are being used to treat at-risk ARMS patients [34]. Due to the shortage of efficacious treatment options, there have been insufficient advancements when it comes to treatment options, thus hindering improvement in the outcome of metastatic or relapsed ARMS [35]. Vincristine, actinomycin D, and cyclophosphamide (VAC), administered intravenously, are considered the classic chemotherapy drugs in North America, with only the duration and dosage of each being modified over the last four decades [34]. Multiple studies have been conducted to improve treatment plans, using a variety of chemotherapeutic agents that target the characteristic features of ARMS, including inhibition of myogenic differentiation. Thus, many of these drug strategies focus on the ability to ‘rehabilitate’ differentiation.

### 3.1. Vincristine, Actinomysin D, and Cyclophosphamide (VAC)

The VAC regimen consisting of the FDA-approved drugs vincristine, actinomycin D, and cyclophosphamide is the standard chemotherapy combination with a response rate of ~70–80% in ARMS patients [36]. Vincristine is an anti-tumor vinca alkaloid considered to be a potent microtubule inhibitor [37,38]. It acts by irreversibly binding to and stabilizing tubulin, thereby interfering with microtubule polymerization, inherently preventing mitosis and inhibiting cell growth [39]. Actinomycin D (ActD) is a known transcription inhibitor; it binds guanine residues and inhibits the function of DNA-dependent RNA polymerase, therefore it prevents RNA synthesis [40]. This characteristic permits it to act as a cytotoxic inducer of apoptosis against tumor cells [40]. A study also showed that when administered in low controlled doses, actinomycin D promotes myogenic differentiation while inhibiting proliferation, thereby making cells less resistant to chemotherapy [41]. Cyclophosphamide is a nitrogen mustard drug that has the ability to alkylate DNA; it does so by metabolizing into its active form phosphoramide [42]. Phosphoramide has the capability of forming cross-linkages at the guanine N-7 position, both in between and within DNA strands, thus leading to programmed cell death [42].

All three of these drugs act differentially to collectively halt tumor growth. This is achieved by either preventing cell division, stopping metastasis, or facilitating cell death, therefore proving to be an effective chemotherapy standard for ARMS patients (Figure 6). 

### 3.2. Cabozantinib (XL184)

As discussed above, in ARMS, MET signaling promotes growth and proliferation, inhibits myogenic differentiation, and increases its metastatic potential [35]. The receptor tyrosine kinase inhibitor, cabozantinib (XL184), upstream of the MET pathway, is used to counteract and impair tumor cell proliferation and angiogenesis, promoting myogenic differentiation [35]. In a phase 1 clinical trial, a total of 41 patients, with a median age of 13 years (range 4–18), were administered cabozantinib to attain a weekly cumulative dosage equivalent to 30 mg/m^2^/day (n = 6), 40 mg/m^2^/day (n = 23), or 55 mg/m^2^/day (n = 12) [43]. At the 40 mg/m^2^/day dosage, dose-limiting toxicities (DLTs) included palmar–plantar erythrodysesthesia syndrome, mucositis, and elevated levels of alanine aminotransferase, lipase, and bilirubin. At 55 mg/m^2^/day, DLTs comprised hypertension, reversible posterior leukoencephalopathy syndrome, headache, fatigue, and proteinuria. Common non-DLTs encompassed diarrhea, hypothyroidism, fatigue, nausea, vomiting, elevated hepatic transaminases, and proteinuria. DLTs were observed across all dosage levels in subsequent cycles. Consistently, steady-state exposure and peak concentrations of cabozantinib remained similar across all dosage levels. Four patients exhibited a confirmed partial response: two with medullary thyroid carcinoma (MTC), one with Wilms tumor, and one with clear cell sarcoma. Stable disease (lasting >6 cycles) was observed in seven patients, including those with MTC, Ewing sarcoma, synovial sarcoma, alveolar soft part sarcoma, paraganglioma, and ependymoma [43]. Considering the toxicity profile, pharmacokinetics, and treatment responses observed, the suggested dose of cabozantinib for pediatric patients with resistant solid tumors is 40 mg/m^2^/day. Currently, a phase 2 trial investigating cabozantinib is underway. The cell signaling mechanism of cabozantinib is summarized in Figure 7.

### 3.3. Bortezomib

Another characteristic feature of ARMS is that it exhibits decreased levels of apoptosis and is successful in evading cell cycle arrest [34]. In vitro studies have shown that when treated with bortezomib, a protease inhibitor that functions by inhibiting the 26S proteosome, ARMS cell lines express increased levels of apoptosis and cell cycle arrest [35].

### 3.4. Vinorelbine

A recent clinical trial was conducted to test the effectiveness of vinorelbine, a second-generation semisynthetic vinca alkaloid with antimitotic and anticancer properties [9]. Vinorelbine acts by inhibiting microtubule dynamics by binding microtubular proteins in the mitotic spindle, preventing chromosomal segregation and triggering the cancerous cells to undergo apoptosis [9]. A recent meta-analysis included five phase 2 trials involving patients with RMS and it evaluated the effect of vinorelbine in these patients [9]. Among these trials, two investigated the efficacy of vinorelbine alone, two examined vinorelbine combined with low-dose oral cyclophosphamide, and one assessed vinorelbine and intravenous cyclophosphamide in conjunction with temsirolimus or bevacizumab. All patients with RMS had either relapsed or refractory disease and had undergone at least one prior treatment. Response to treatment was evaluated based on RECIST1.1 criteria, defining a response as either complete or partial. Response data were sourced from published findings or directly from the principal investigators of the trials. Rhabdomyosarcoma not otherwise specified (RMS NOS) patients were categorized with embryonal rhabdomyosarcoma (ERMS) patients for the purpose of this analysis. Summary estimates comparing the differences in response rates between alveolar RMS (ARMS) and ERMS were generated using a random-effects model to accommodate the heterogeneity observed among the studies [9]. The meta-analysis included a total of 156 enrolled patients who were assessed for response, consisting of 85 with alveolar rhabdomyosarcoma (ARMS), 64 with embryonal rhabdomyosarcoma (ERMS), and 7 with rhabdomyosarcoma not otherwise specified (RMS-NOS). The combined analysis using a random-effects model revealed a 41% higher response rate (*p* = 0.001, 95% CI; 0.21–0.60) to vinorelbine as a single agent or in combination therapy in ARMS patients, compared to those with ERMS. There was no significant difference observed in the rate of disease progression between ARMS and ERMS patients (*p* = 0.1, 95% CI; −0.26–0.02) [9]. The recommended combination therapy is a low dose of cyclophosphamide, ~25 mg/m^2^ per day for 28 days, with the administration of 25 mg/m^2^ of vinorelbine on days 1, 8, and 15 [44].

### 3.5. AZD1775

Wee1 kinase is a cell cycle regulator and it acts by inhibiting cyclin-dependent kinase 1 and phosphorylation [35]. AZD1775 is a selective tyrosine kinase inhibitor that has been reported to inhibit the growth of several sarcoma cell lines; however, its role in ARMS is still misunderstood [35].

A recent study illustrated that AZD1775 could serve as a viable therapeutic agent in combination with conventional chemotherapy [35]. The Wee1 kinase is a cell cycle regulator that specifically maintains the cell in the G2/M phase, thereby providing sufficient time to repair DNA prior to undergoing mitosis (Kahen et al., 2016). When inhibited by AZD1775, however, the CDK1/2 activity takes over unchecked, allowing the cells to prematurely progress through the G2/M phase and undergo mitosis. The outcome of this leads to DNA strand breaks, mitotic dysfunction, and cell death [35].

Previous literature has also shown that when used in combination with gemcitabine, AZD1775 leads not only to a delay in tumor growth, but also to smaller tumors in osteosarcoma cells [35]. It is also hypothesized that if combined with conventional chemotherapy, they could work synergistically to make the DNA damage more potent [35]. These findings indicate the potential of this agent for improved treatment strategies for ARMS, once evaluated more in future studies [35].

### 3.6. Entinostat, Panobinostat, and Vorinostat

Histone deacetylase (HDAC) is an epigenetic marker that when inhibited has antitumor effects, as has been demonstrated using an RMS model [34]. Agents such as entinostat, panobinostat, and vorinostat are known HDAC inhibitors and have been reported to delay tumor growth in RMS xenografts [34]. In addition to disrupting transcriptional complexes, leading to suppression of key oncogenic genes, HDAC inhibitors also have the capability to induce transcriptional stress, resulting in their terminal differentiation or apoptotic cell death [34] (Figure 8).

### 3.7. Critotinib, Bevacizumab (mAb), and Regorafenib

The constitutive activation of receptor tyrosine kinases (RTKs) such as ALK, MET, and VEGFR is known to promote tumor progression in ARMS, by reprogramming many intracellular pathways, such as those involved in differentiation [34]. Currently, there are two main strategies for targeting RTKs: small molecule kinase inhibitors and immunotherapy-like monoclonal antibodies. Critotinib, bevacizumab (mAb), and regorafenib are a few examples of RTK inhibitors that have successfully induced tumor regression in pre-clinical models [34].

### 3.8. All-Trans Retinoic Acid (ATRA)

All-trans retinoic acid (ATRA) is an important metabolite of vitamin A, mediating the functions of growth and development [45]. It plays a crucial role in cell proliferation, cell differentiation, apoptosis, and embryonic development [46]. ATRA primarily exerts its effects through its interactions with nuclear retinoic acid receptors (RARs), which are transcription factors [47]. RARs form heterodimers with retinoid X receptors (RXRs) and regulate the transcription of target genes [48,49]. The RAR signaling pathway plays a crucial role in mediating the differentiation-promoting effects of ATRA in ARMS [50]. By activating RARs, ATRA can modulate the expression of genes associated with myogenic differentiation, the process by which ARMS cells transform into more mature, muscle-like cells.

ATRA exerts differentiation-inducing effects in ARMS through various mechanisms. For instance, it can act by upregulating MYOD1, a crucial myogenic regulator, promoting commitment to the muscle lineage, and facilitating ARMS cell differentiation [51]. Moreover, ATRA also targets the hallmark genetic abnormality in ARMS, inhibiting the PAX3-FOXO1 fusion protein’s expression and activity, thus disrupting oncogenic signaling and facilitating differentiation [49]. Finally, ATRA induces epigenetic changes, such as DNA methylation and histone modifications, influencing the expression of genes related to myogenic differentiation in ARMS cells [52]. These mechanisms collectively contribute to ATRA’s potential as a therapeutic agent for ARMS differentiation.

### 3.9. Cisplatin

Cisplatin is a toxic antineoplastic agent, but one of the most heavily employed agents that first came to use in the 1970s. It operates by inducing DNA damage and subsequent cell cycle arrest, rather than directly promoting differentiation in cancer cells [53]. It forms covalent bonds with DNA, leading to the formation of DNA crosslinks and adducts, triggering DNA repair responses and, in ARMS cells, potentially causing apoptosis rather than differentiation [54]. Cisplatin is often administered in combination with other chemotherapeutic agents to enhance treatment efficacy and inhibit tumor growth. Additionally, it can act as a radiosensitizer, increasing the sensitivity of cancer cells to radiation therapy, which is commonly part of ARMS treatment [55]. The utilization of various therapeutic approaches, including cisplatin, aims to achieve comprehensive control of ARMS, with treatment plans tailored to individual patients in consultation with medical oncologists.

### 3.10. 5-Azacytidine

5-Azacytidine is a demethylating agent which functions by incorporating itself into DNA during replication and inhibiting DNA methyltransferase activity, resulting in DNA demethylation [56]. In ARMS, hypermethylation of specific genes can lead to their silencing, including those associated with differentiation [57]. 5-Azacytidine treatment can reverse this process, allowing previously silenced genes linked to myogenic differentiation to be re-expressed. This demethylation also contributes to the restoration of myogenic regulatory factors (MRFs) such as MyoD and myogenin, which are crucial for myogenic differentiation [58]. Furthermore, 5-Azacytidine can induce cell cycle arrest and cellular senescence, promoting a more mature cellular state in ARMS [59]. However, it is important to note that 5-Azacytidine’s effectiveness often varies among patients and is often used in the context of personalized treatment plans for ARMS, typically under clinical trial conditions.

Table 1 summarizes the key molecular targets of the drugs discussed.

## 4. In Vivo Model for RMS Investigations

There are several pre-clinical models to investigate new therapeutic approaches to improve RMS patient health. Both mouse and zebrafish models are among the most popular for pre-clinical investigations [55]. Broadly speaking, there are four primary categories of mouse models employed in rhabdomyosarcoma (RMS) research: (1) cell-line-derived xenografts (CDXs); (2) patient-derived xenografts (PDXs); (3) environmentally induced mouse models (EIMMs); and (4) genetically engineered mouse models (GEMMs). CDXs involve implanting specific cell lines subcutaneously into immunocompromised mice to generate models closely resembling human tumors, categorized as orthotopic or heterotopic models. These models are frequently utilized in pediatric RMS investigations. PDXs entail injecting primary tumor tissue subcutaneously into immunocompromised mice to obtain either cells or tissue fragments, again classified as orthotopic or heterotopic. GEMMs involve introducing specific genetic alterations, often from an oncogene or tumor suppressor gene, to create the model. Different types of germline or somatic mutations are employed, depending on the research objectives, to generate various types of GEMMs. Lastly, EIMMs involve exposing animals to mutagens such as oxidative stress, aging, or DNA methylation to simulate disease progression, providing diverse insights into mutagenesis processes [55].

In addition to mouse models, zebrafish models have more recently been utilized to examine a range of rhabdomyosarcoma (RMS)-related features specific to development, histology, pathogenesis, tumor progression, metastasis, and drug screening. Given that embryonal RMS (ERMS) is the predominant subtype in humans, and that the biological and clinical similarities between fusion gene-negative alveolar RMS (ARMS) and ERMS have been established, initial zebrafish models have primarily focused on the ERMS subtype. ERMS is characterized by mutations in genes encoding RAS GTPases, MYOD1, and FGFR4. Conversely, in fusion-positive RMS, the overexpression of the fusion protein PAX3/PAX7-FOXO1, resulting from chromosomal translocation, leads to a more aggressive form of RMS or accounts for approximately 85% of ARMS cases. These two types of RMS models can be recapitulated in zebrafish through either genetic modifications or the transplantation of tumor cells [60,61,62].

## 5. Conclusions and Future Direction

Targeting tumor cells using multiple targeted strategies is becoming a more popular approach to providing chemotherapy. As an example, in addition to the current therapies, there are ongoing investigations exploring if simultaneous targeting of differentiation pathways and other chemotherapy agents could be more effective in ARMS therapy. Since ARMS cells show a decrease in myogenic differentiation, this is becoming a major focus of future studies. As mentioned above, the TGF-β pathway could be an important research target for better chemotherapeutic outputs in ARMS because TGF-β has been observed to be involved in decreased differentiation in ARMS tumor cells. Future studies should focus on multiple molecularly targeted therapies since the addition of cytotoxic agents to the current chemotherapy regime has depicted minimal success [5].

Functionalized nano-carriers, a product of multidisciplinary collaboration across medicine, pharmacy, material science, and engineering, have emerged as an ideal option for implementing combination therapy [63]. Their numerous advantages include passive targeting capacity via enhanced permeation and retention, a high surface-to-volume ratio, the capability to accommodate various drugs, and adjustable surfaces for targeting modifications [63,64]. At the turn of the century, nano-sized particles that had been clinically approved for treating other solid tumors were investigated for use in sarcomas [65]. However, the outcomes were mostly anecdotal, and the clinical benefits were not significant. Recently, a new nanosystem formulation called NBTXR3 has entered phase 2–3 trials for sarcoma treatment [65]. Initial results are promising and could pave the way for further research in nanotechnology. On the other hand, there is growing interest in exploring innovative high-throughput approaches that integrate microfluidics, bioengineering, nanotechnology, and cell or tissue utilization to create lab-on-a-chip devices. These platforms hold promise for applications in diagnosis, prognosis, and drug screening [66]. Recently, integrin receptor-targeted Lipid–Protamine-siRNA (LPR) nanoparticles utilizing the RGD peptide were developed and their target specificity, as well as their post-silencing effects, were validated. The researchers demonstrated that RGD-LPRs exhibit specificity to ARMS both in vitro and in vivo. When loaded with siRNA targeting the P3F breakpoint, these nanoparticles effectively downregulated the fusion transcript and suppressed cell proliferation, albeit without inducing significant apoptosis. In a xenograft ARMS model, P3F-targeting LPR nanoparticles demonstrated a statistically significant delay in tumor growth and inhibition of tumor initiation when administered concomitantly with tumor cells [67]. Moreover, our team has recently designed a specific ATRA encapsulated in bioengineered MESH material [68] that could be used in the future for combination targeted therapy for ARMS. Bearing in mind the ground-breaking findings that point toward an overexpression of TGF-β in ARMS, it is safe to say that future research should invest greater time in understanding this pathway and its role in ARMS. Developing new treatment therapies that specifically target TGF-β and its effect on ARMS differentiation and lead to its downregulation would present a promising advancement in improving the outcome of ARMS patients.

## Figures and Tables

**Figure 1 ijms-25-02791-f001:**
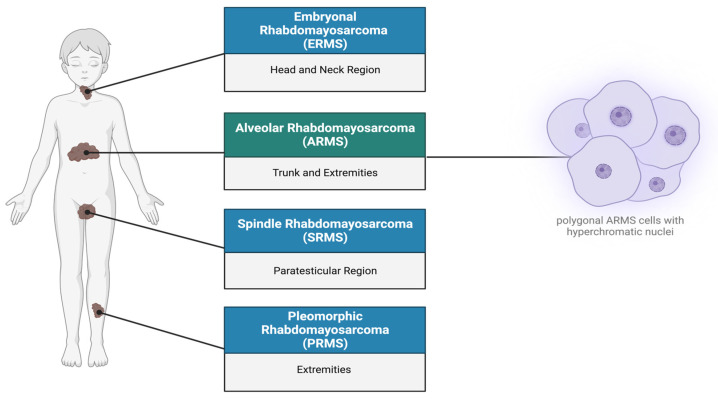
Subtypes of rhabdomyosarcoma. Rhabdomyosarcoma can be differentiated into four subtypes: (1) Embryonal rhabdomyosarcoma occurs primarily in the head, neck, and bladder regions. (2) Alveolar rhabdomyosarcoma (ARMS) is predominantly found in the trunk, arms, and legs, characterized by polygonal cells with hyperchromatic nuclei. (3) Spindle rhabdomyosarcoma (SRMS) occurs in the para testicular, head, and neck regions. (4) Pleomorphic rhabdomyosarcoma (PRMS) is localized in the deep soft tissues of the body extremities. Created with BioRender.com, accessed on 22 June 2023.

**Figure 2 ijms-25-02791-f002:**
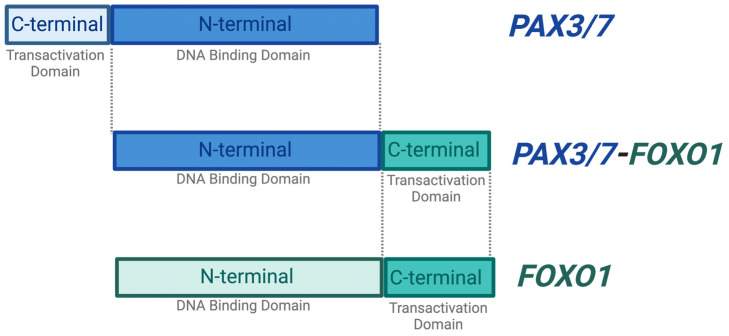
*PAX3/7-FOXO1* fusion gene. The majority of ARMS cases are caused by a chromosomal translocation event between the N-terminal DNA binding domain of the *PAX3* or *PAX7* and the C-terminal transactivation domain of the *FOXO*1 gene. These events result in a fusion gene *PAX3/7-FOXO1*, which is found to be an oncogenic driver. The PAX3/7-FOXO1 fusion gene retains the DNA binding activity of *PAX3/7* and is therefore capable of initiating transcription. Created with BioRender.com, accessed on 27 June 2023.

**Figure 3 ijms-25-02791-f003:**
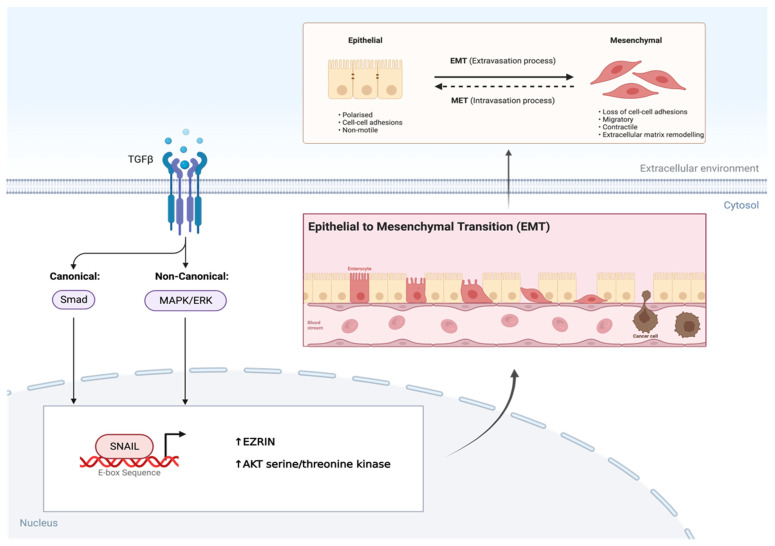
TGF-β induces EMT via SNAIL transcription factor in AMRS. Overexpression of TGF-β in AMRS leads to phosphorylation and activation of its two major downstream pathways. This leads to SNAIL, a transcription factor binding to the E-box sequence in the promotor region of genes, resulting in the epithelial to mesenchymal transition (EMT). Created with BioRender.com, accessed on 28 June 2023.

**Figure 4 ijms-25-02791-f004:**
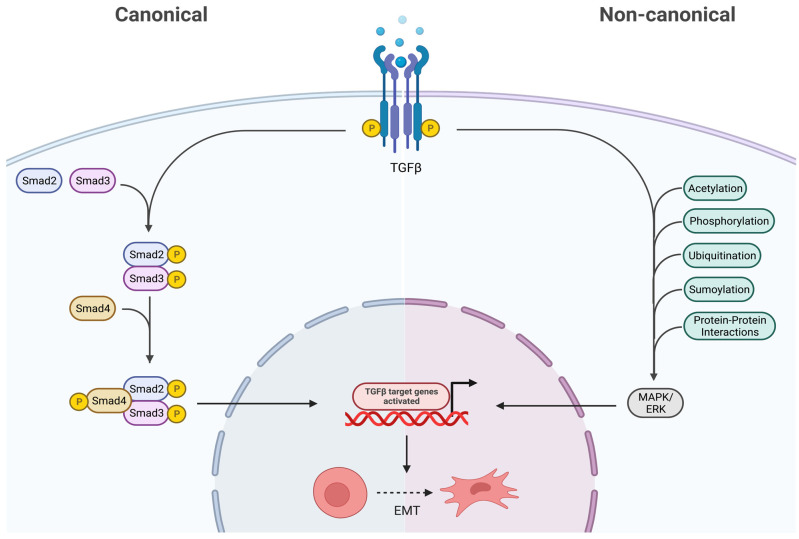
Canonical and non-canonical downstream pathways of TGF-β. In the canonical pathway, the SMAD 2/3 are phosphorylated via TGF-β, thus translocating into the nucleus and regulating gene transcription. The non-canonical pathway activates downstream cascades like MAPK/ERK via various post-translational modifications. Both these pathways ultimately result in epithelial to mesenchymal transition (EMT), therefore they promote cancer. Created with BioRender.com, accessed on 28 June 2023.

**Figure 5 ijms-25-02791-f005:**
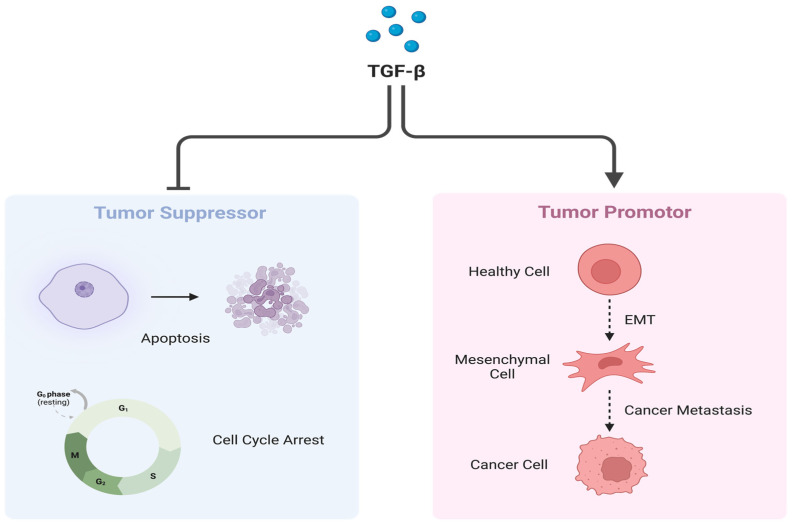
Dual role of TGF-β. TGF-β under normal circumstances plays a crucial role as a tumor suppressor by regulating cell cycle arrest and inducing apoptosis (**Left**). However, when overexpressed, TGF-β promotes cancer progression and metastasis via EMT induction (**Right**). Created with BioRender.com, accessed on 22 June 2023.

**Figure 6 ijms-25-02791-f006:**
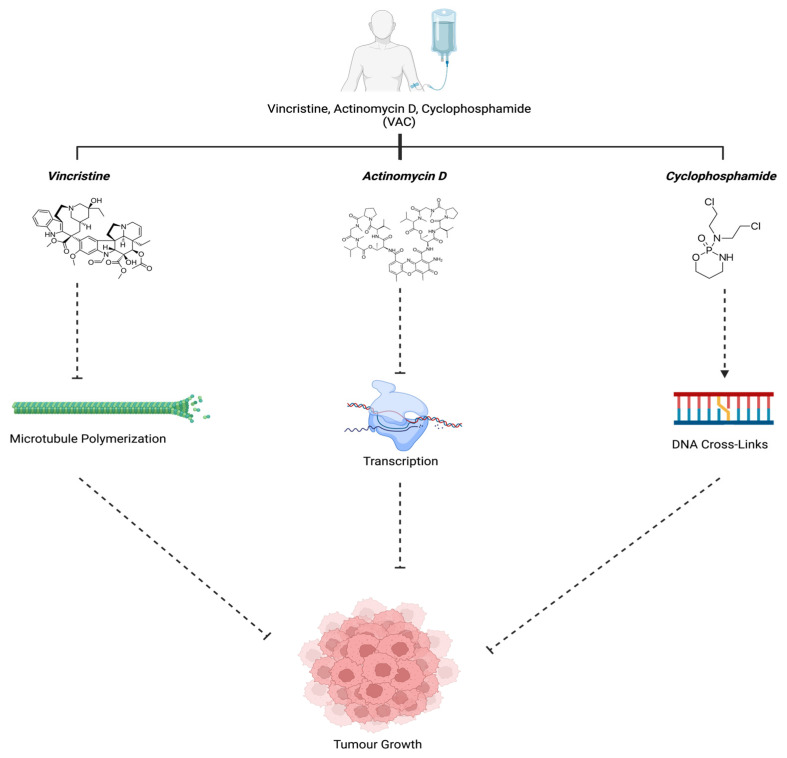
VAC mechanism of action. Vincristine (Group6-3, 2011), Actinomycin D (*Actinomycin D*, n.d.), and Cyclophosphamide (“Cyclophosphamide”, 2023) work in unison to prevent tumor growth. Created with BioRender.com, accessed on 12 July 2023.

**Figure 7 ijms-25-02791-f007:**
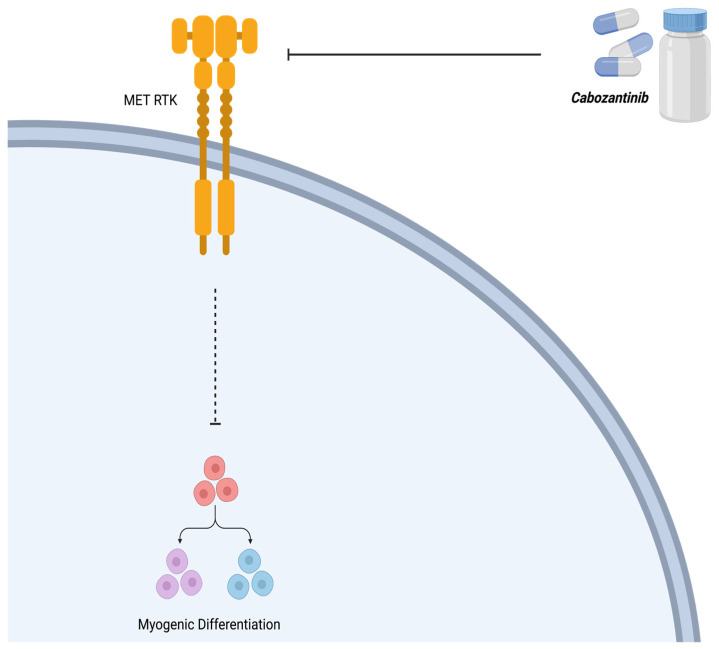
Mechanism of action of cabozantinib (XL184). Cabozantinib is a receptor tyrosine kinase (RTK) inhibitor that counteracts the effect of MET, thereby allowing myogenic differentiation. Created with BioRender.com, accessed on 12 July 2023.

**Figure 8 ijms-25-02791-f008:**
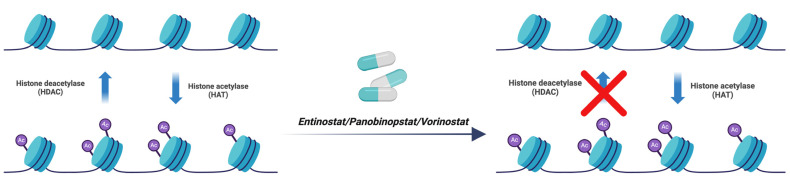
Mechanism of action of inhibitors of histone deacetylase (HDAC). Entinostat, panobinostat, and vorinostat are agents that inhibit histone deacetylation, resulting in delayed tumor growth and apoptosis. (red X means blocking). Created with BioRender.com, accessed on 22 June 2023.

**Table 1 ijms-25-02791-t001:** Chemotherapy drugs used for ARMS and their molecular targets.

Drug/Compound	Molecular Target	References
Vincristine, Actinomycin D, and Cyclophosphamide (VAC)	Microtubule Polymerization, Guanine Nucleotide in DNA, and Cross-Linkages with Guanine N-7, respectively	[40,42]
Cabozantinib (XL184)	Tyrosine Kinase (MET)	[35]
Bortezomib	26s Proteosome	[35]
Vinorelbin	Microtubular Proteins	[9]
AZD1775	Wee1	[35]
Entinostat, Panobinostat, and Vorinostat	Histone Deacetylase (HDAC)	[34]
Critotinib, Bevacizumab (mAb), and Regorafenib	Receptor Tyrosine Kinae (RTK)	[34]
All-Trans Retinoic Acid (ATRA)	Retinoic Acid Receptors (RARs)	[20]
Cisplatin	DNA	[53]
5-Azacytidine	DNA Methyltransferase	[56]

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
