# Peer review of "Transforming Growth Factor Beta and Alveolar Rhabdomyosarcoma: A Challenge of Tumor Differentiation and Chemotherapy Response"

_ijms, 2024, doi:10.3390/ijms25052791_

Round 1

Reviewer 1 Report

Comments and Suggestions for Authors

The manuscript focuses on Transforming Growth Factor Beta and Alveolar Rhabdomyosar coma: A Challenge of Tumor Differentiation and Chemother apy Response represents a technically correct and timely relevant manuscript able to be published on this journal after minor suggestions:

- In the manuscript, please, could the authors improve the clinical readability of cited clinical approaches? In my opinion, this point may be considered an interesting challenging point

- In the manuscriopt, please, could the authors also add a brief description of methodological  techniques able to investigate these promising biomakers?

- In the conclusion, please, could the authors consider how this approach may be helpful in the clinical management of alveolar rhabdomyosarcoma patients?

Comments on the Quality of English Language

Minor english revision

Author Response

- In the manuscript, please, could the authors improve the clinical readability of cited clinical approaches? In my opinion, this point may be considered an interesting challenging point

Answer: The authors appreciate the important comment by the respected reviewer. We explain in detail the clinical trials as following:  

  1. Page 7-8, Line: 198-216 (Cabozantinib (XL184)).
  2. Page 8-9, Line: 231-259 (Vinorelbine): The original paper should be read carefully and 150 words about the clinical trial that has been mentioned in Ref 9 should be written. (done).

- In the manuscript, please, could the authors also add a brief description of methodological  techniques able to investigate these promising biomakers?

Answer: The authors have followed the respected reviewer comment and added a section entitled (In Vivo Model for RMS Investigations) to explain the in vivo model to investigate new therapies for different types of RMS (339-368).

- In the conclusion, please, could the authors consider how this approach may be helpful in the clinical management of alveolar rhabdomyosarcoma patients?

Answer: The authors have added (In Vivo Model for RMS Investigations) to explain the in vivo model to investigate new therapies for different types of RMS (339-368) and also added a section in the future direction the impact of nano-technology in treatment of different types of sarcoma including ARMS (line 381-402).

Reviewer 2 Report

Comments and Suggestions for Authors

The authors of this manuscript performed a literature review about the role of TGFbeta in rhabdomyosarcoma, focusing on the alveolar subtype. They also  described the main therapeutic options, briefly discussing also future directions. 

The manuscript is well written and organised, however some suggestions are provided in order to improve the overall quality:

1) when discussing future directions, authors mention bioengineered materials as promising option to improve therapeutic options. This is a particularly relevant topic, since nanotechnology in last years has made great advancements (10.3389/fbioe.2022.953555, 10.3109/1061186X.2011.622399, 10.1016/j.jconrel.2016.05.063). In this regard, it would be nice to further expand this topic adding a specific paragraph on the potential role of nanomedicine in sarcomas and rhabdomyosarcoma as well, including the mentioned references. 

Comments on the Quality of English Language

English is fine, only minor spelling mistakes are present

Author Response

  • when discussing future directions, authors mention bioengineered materials as promising option to improve therapeutic options. This is a particularly relevant topic, since nanotechnology in last years has made great advancements (10.3389/fbioe.2022.953555, 10.3109/1061186X.2011.622399, 10.1016/j.jconrel.2016.05.063). In this regard, it would be nice to further expand this topic adding a specific paragraph on the potential role of nanomedicine in sarcomas and rhabdomyosarcoma as well, including the mentioned references. 1) when discussing future directions, authors mention bioengineered materials as promising option to improve therapeutic options. This is a particularly relevant topic, since nanotechnology in last years has made great advancements (10.3389/fbioe.2022.953555, 10.3109/1061186X.2011.622399, 10.1016/j.jconrel.2016.05.063). In this regard, it would be nice to further expand this topic adding a specific paragraph on the potential role of nanomedicine in sarcomas and rhabdomyosarcoma as well, including the mentioned references. 

Answer: The authors appreciate the constructive comment of the respected reviewer. We added a section in the future direction sections and discussed the strategies using nano carrier for cancer treatment in general and then added few lines about the impact of nano carriers in treatment of sarcoma .(line 381-402). We also used the articles that were suggested by the respected reviewer and added them to the literature of the revised paper.